# NLRP1 restricts butyrate producing commensals to exacerbate inflammatory bowel disease

Hazel Tye[1,2], Chien-Hsiung Yu[1,2], Lisa A. Simms[3], Marcel R. de Zoete[4,5], Man Lyang Kim[1,2], Martha Zakrzewski[6], Jocelyn S. Penington [7], Cassandra R. Harapas[1,2], Fernando Souza-Fonseca-Guimaraes[2,8], Leesa F. Wockner [9], Adele Preaudet[1,2], Lisa A. Mielke[2,8,10], Stephen A. Wilcox[2,11], Yasunori Ogura[12], Sinead C. Corr[13], Komal Kanojia[14], Konstantinos A. Kouremenos[14], David P. De Souza[14], Malcolm J. McConville[14,15], Richard A. Flavell[4,16], Motti Gerlic [17], Benjamin T. Kile[2,18,19], Anthony T. Papenfuss [2,7,20,21], Tracy L. Putoczki[1,2], Graham L. Radford-Smith [3,22,23] & Seth L. Masters [1,2]

Anti-microbial signaling pathways are normally triggered by innate immune receptors when detecting pathogenic microbes to provide protective immunity. Here we show that the inflammasome sensor Nlrp1 aggravates DSS-induced experimental mouse colitis by limiting beneficial, butyrate-producing *Clostridiales* in the gut. The colitis-protective effects of *Nlrp1* deficiency are thus reversed by vancomycin treatment, but recapitulated with butyrate supplementation in wild-type mice. Moreover, an activating mutation in *Nlrp1a* increases IL-18 and IFNγ production, and decreases colonic butyrate to exacerbate colitis. We also show that, in patients with ulcerative colitis, increased *NLRP1* in inflamed regions of the colon is associated with increased *IFN-γ*. In this context, *NLRP1*, *IL-18* or *IFN-γ* expression negatively correlates with the abundance of *Clostridiales* in human rectal mucosal biopsies. Our data identify the NLRP1 inflammasome to be a key negative regulator of protective, butyrate-producing commensals, which therefore promotes inflammatory bowel disease.

[1] Inflammation Division, The Walter and Eliza Hall Institute of Medical Research, Parkville, VIC 3052, Australia. [2] Department of Medical Biology, University of Melbourne, Parkville, VIC 3010, Australia. [3] Gut Health, QIMR Berghofer Medical Research Institute, Brisbane 4029 QLD, Australia. [4] Department of Immunobiology, Yale University School of Medicine, New Haven, CT 06519, USA. [5] Department of Infectious Diseases and Immunology, Utrecht University, Utrecht 3584 CL, The Netherlands. [6] Medical Genomics, QIMR Berghofer Medical Research Institute, Brisbane 4029 QLD, Australia. [7] Bioinformatics Division, The Walter and Eliza Hall Institute of Medical Research, Parkville, VIC 3052, Australia. [8] Molecular Immunology Division, The Walter and Eliza Hall Institute of Medical Research, Parkville, VIC 3052, Australia. [9] Statistics Division, QIMR Berghofer Medical Research Institute, Brisbane 4029 QLD, Australia. [10] Olivia Newton-John Cancer Research Institute, School of Cancer Medicine, La Trobe University, Heidelberg, VIC 3084, Australia. [11] Systems Biology and Personalized Medicine Division, The Walter and Eliza Hall Institute of Medical Research, Parkville, VIC 3052, Australia. [12] Department of Food Science and Nutrition, Nara Women's University, Nara 6308506, Japan. [13] Department of Microbiology, The Moyne Institute of Preventative Medicine, School of Genetics and Microbiology, Trinity College Dublin, Dublin 2, Ireland. [14] Metabolomics Australia, Bio21 Institute of Molecular Science and Biotechnology, University of Melbourne, Parkville, VIC 3010, Australia. [15] Department of Biochemistry and Molecular Biology, Parkville, VIC 3010, Australia. [16] Howard Hughes Medical Institute, Yale University, New Haven, CT 06510, USA. [17] Department of Clinical Microbiology and Immunology, Sackler Faculty of Medicine, Tel Aviv University, Tel Aviv 69978, Israel. [18] ACRF Chemical Biology Division, The Walter and Eliza Hall Institute of Medical Research, Parkville, VIC 3052, Australia. [19] Department of Anatomy and Developmental Biology, Monash Biomedicine Discovery Institute, Monash University, Clayton 3800 VIC, Australia. [20] Sir Peter MacCallum Department of Oncology, University of Melbourne, Parkville, VIC 3010, Australia. [21] Bioinformatics and Cancer Genomics Lab, Peter MacCallum Cancer Centre, Melbourne, VIC 3002, Australia. [22] Department of Gastroenterology, Royal Brisbane and Women's Hospital, Brisbane 4029 QLD, Australia. [23] University of Queensland School of Medicine, Brisbane 4029 QLD, Australia. These authors contributed equally: Hazel Tye, Chien-Hsiung Yu. Correspondence and requests for materials should be addressed to S.L.M. (email: masters@wehi.edu.au)

nflammatory bowel disease (IBD) predominantly affects people in westernized countries[1], and includes both Crohn's disease (CD) and ulcerative colitis (UC). Since the underlying cause of IBD is not well understood, there is a need to understand the molecular mechanisms that drive pathogenesis in order to improve therapeutic options and patient quality of life. Two inflammatory cytokines central to the pathogenesis of IBD are interleukin (IL)-1β and IL-18. IL-1β is increased in the mucosa of patients with IBD and is associated with the proliferation of pathogenic T helper 17 (Th17) cells[2]. Despite this, in acute models of IBD, deletion of IL-1β has been shown to contribute to defective repair of the epithelial barrier, resulting in increased disease[3]. Regarding IL-18, increased levels were detected in the serum of IBD patients compared to healthy controls[4]. IL-18 has been shown to enhance interferon-γ (IFNγ) production in T cells[5], with a genetic association between a single-nucleotide polymorphism in IFNG and increased severity of CD and UC[6]. Collectively, this would suggest that the effect of IL-18 on IFNγ-producing Th1 cells may be pathogenic in IBD. However, the genetic deletion of IL-18 from mice has opposing effects in models of colitis, depending on whether it is produced by hematopoietic or non-hematopoietic cells[7–9]. Therefore, both IL-1β and IL-18 may have pleiotropic effects in IBD, perhaps depending on localization, disease severity, kinetics of cytokine production or other factors such as microbiome colonization and species differences.

IL-1β and IL-18 are both activated by cleavage, via an intracellular complex of proteins containing Caspase-1, known as the inflammasome. Inflammasome complexes are nucleated by an innate immune receptor, including members of the Nod-Like Receptor (NLR) family. The last decade has seen a dramatic increase in research regarding the role of the microbiome and inflammasomes in IBD[7,10]. This link was initially established with the finding that severe dextran sulfate sodium (DSS)-induced colitis observed in NLRP3- and NLRP6-deficient mice could be transferred to co-housed wild-type (WT) mice[7,10]. More recently, this pathway has been carefully re-examined through the analysis of littermate control mice, and the conclusion reached was that NLRP6, and indeed mice deficient for the inflammasome adaptor ASC (apoptosis-associated speck like protein containing a caspase recruitment domain), had no microbial dysbiosis or DSS-colitis phenotype[11]. In spite of this, Caspase-1-deficient mice, even with a normalized microbiome, exhibit protection against DSS-colitis[12], suggesting a possible role for ASC-independent inflammasomes regulating IL-1β or IL-18 in IBD.

At least in mice, Nlrp1a can form an ASC-independent inflammasome, as demonstrated by an activating mutation (Nlrp1a^Q593P/Q593P) which results in autoinflammatory disease that is rescued by the genetic deletion of Caspase-1, but not ASC[13]. Unlike humans, mice have three paralogs of the Nlrp1 gene (Nlrp1a, b, c)[14] and while studies have shown anthrax lethal toxin activates mouse Nlrp1b in macrophages[15], this does not hold true for human NLRP1[14]. Since Nlrp1c is a pseudogene, it may be that Nlrp1a is a closer functional homolog of human NLRP1. Indeed, deletion of the N-terminal domain of Nlrp1a renders the molecule auto-activated[16], similar to loss-of-function mutations in the N-terminal domain of human NLRP1[17]. These rare loss-of-function mutations in the N-terminal domain of NLRP1 result in a familial autoinflammatory skin disease associated with cancer. Additionally, rare mutations in a linker region between the NACHT and LRR domains can cause a similar disease[18], which is the same location as we observed for the mutation activating mouse Nlrp1a[13].

Aside from familial mutations in NLRP1 that predispose humans to skin cancer, common polymorphisms at the NLRP1 locus are associated with resistance to glucocorticoid treatment in

pediatric IBD, and several autoimmune diseases such as vitiligo, celiac disease and psoriasis[15,19–22]. NLRP1 polymorphisms are also strongly associated with skin extra-intestinal manifestations in CD[23]. NLRP1 is expressed by a variety of cell types, which are predominantly hematopoietic; however, expression is also seen within glandular epithelial structures including the lining of the small intestine, stomach and colon[24].

Given these links between IBD and NLRP1, we use the model of DSS-induced colitis in mice that are deficient for all three paralogs of Nlrp1 (Nlrp1^−/−) and find that they are protected from severe disease pathology. Mice with single deletion of Nlrp1a are also protected while conversely, mice with an activating mutation in Nlrp1a suffer more severe disease that can be resolved by the genetic deletion of IL-18. Moreover, we demonstrate that increased IL-18 is associated with an increased Th1 response during DSS-induced colitis, while loss of Nlrp1 prevents this, and leads to increased butyrate-producing commensals from the Clostridiales order. These data agree with the increased expression of NLRP1 we observe in biopsies from patients with UC, which is correlated with increased IFNγ expression and decreased butyrate-producing Clostridiales.

## Results

**Loss of Nlrp1 suppresses DSS-induced colitis.** In order to determine the role of NLRP1 in acute colitis we administered Nlrp1^−/− mice with 3% (w/v) DSS ad libitum in their drinking water for 6 days. As a comparison, we used WT mice that were the F1 or F2 progeny of Nlrp1^+/− matings, bred in the same facility. Throughout the course of DSS-induced mucosal injury, we observed that the Nlrp1^−/− mice were protected from the clinical features of colitis, as indicated by reduced weight loss, sustained colon length and lower histology score compared to WT mice (Fig. 1a–d). We also studied mice with single deletion of Nlrp1a, and found a similar protection from DSS-induced colitis associated with reduced weight loss and decreased severity of inflammation quantified by colonoscopy (Supplementary Fig. 1a, b). Serum cytokines were quantified at day 7, revealing that IL-18 was decreased in mice lacking Nlrp1a, and explant tissue from the colon of these mice also produced less IL-18 and IFNγ ex vivo (Supplementary Fig. 1c-e). In order to identify whether NLRP1 activity in the non-hematopoietic or hematopoietic compartment was detrimental during DSS-induced colitis, we generated reciprocal bone marrow chimeras, whereby WT and Nlrp1^−/− mice were lethally irradiated and reconstituted with either WT or Nlrp1^−/− bone marrow. At 12 weeks after reconstitution, mice were challenged with DSS for 6 days. A gradient of disease severity was observed with Nlrp1^−/− mice receiving Nlrp1^−/− bone marrow exhibiting the least weight loss, shortening of the colon and histology score (Fig. 1e–g). This was followed by Nlrp1^−/− mice receiving WT bone marrow, then WT mice receiving Nlrp1^−/− bone marrow, and finally WT mice receiving WT bone marrow, which were the most severely affected, as expected. These results indicate that NLRP1 activity from both the hematopoietic and non-hematopoietic compartment influence the outcome of DSS-induced colitis, with a predominant role of Nlrp1 from within the non-hematopoietic compartment.

**Reduced Th1 response for Nlrp1^−/− mice during DSS-colitis.** Since inflammasomes regulate IL-1β and IL-18 production, which are implicated in Th17 and Th1 responses respectively, we employed fluorescence-activated cell sorting (FACS) analysis to assess the downstream consequences of the loss of NLRP1 on Th1 and Th17 populations. Following DSS treatment, the reduced disease severity observed in the Nlrp1^−/− mice was associated with a significant decrease in the total number of CD4+ T cells in

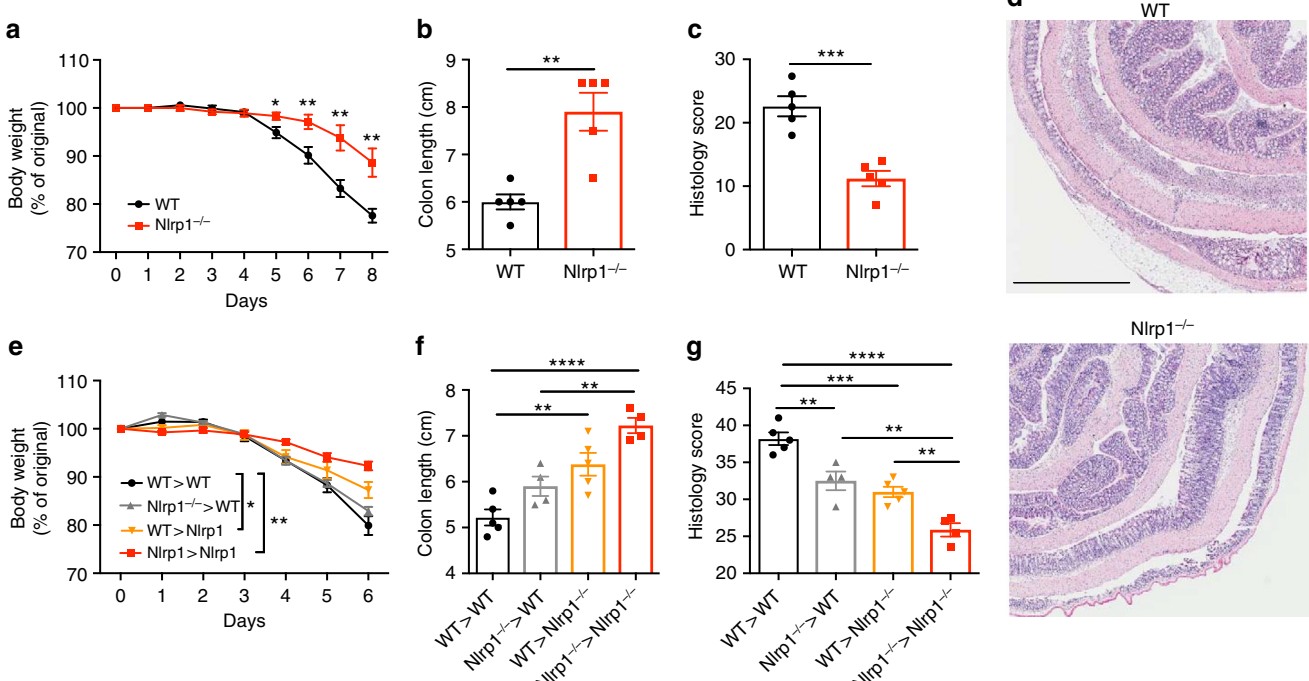

**Fig. 1** Loss of NLRP1 confers protection against DSS-induced colitis. WT and *Nlrp1*$^{-/-}$ mice were given 3% (w/v) DSS for 6 days followed by normal drinking water for 2 days and disease severity was measured according to **a** percentage weight loss, **b** colon length and **c** histology score for epithelial damage and inflammatory cell infiltrate of hematoxylin and eosin (H&E)-stained sections of the colon, scale bar = 1 mm. **d** Representative H&E-stained sections of the colon from WT and *Nlrp1*$^{-/-}$ mice. Lethally irradiated WT and *Nlrp1*$^{-/-}$ mice were reconstituted with WT or *Nlrp1*$^{-/-}$ bone marrow for 12 weeks and were given 3% DSS for 6 days. Disease severity was measured by **e** percentage weight loss, **f** colon length and **g** histology score. Data are representative of 3 independent experiments with 3–5 mice per group. Means ± SEM; *$p \leq 0.05$, **$p \leq 0.01$, ***$p \leq 0.001$, ****$p \leq 0.0001$. As determined by a two-tailed, unpaired *t*-test. A one-way analysis of variance (ANOVA) and Tukey's post-hoc comparisons were performed on data that involved more than two comparisons

the colonic lamina propria (cLP) (Supplementary Fig. 2a). At steady state (SS) no differences in the frequency of IFNγ+ (Th1), IFNγ+IL-17a+ (Th1/Th17) or IL-17a+ (Th17) cells were observed in the cLP or spleen of both WT and *Nlrp1*$^{-/-}$ mice (Supplementary Fig. 2b-d). However, a significant reduction in the frequency of IFNγ-producing CD4+ T cells was observed in the cLP and spleen of DSS-treated *Nlrp1*$^{-/-}$ mice (Supplementary Fig. 2b-d). No significant differences in IFNγ+IL-17a+ and IL-17a+CD4+ T cells were observed between the genotypes after DSS treatment. This is consistent with our results above, in that *Nlrp1*$^{-/-}$ mice display a reduced inflammatory capacity that is predominantly associated with decreased Th1 effector T cells in this mouse model of IBD.

**Nlrp1$^{-/-}$ microbiome can be transferred to co-housed mice**. As an initial test to see if the DSS-colitis phenotype of Nlrp1-deficient mice was related to a commensal imbalance, we performed 16S ribosomal RNA sequencing on stool from littermates of Nlrp1$^{+/-}$ matings, which had been housed individually for 6 weeks after weaning. This revealed that in the period of time since weaning, a significant difference in the abundance of several bacterial operational taxonomic units (OTUs) was observed for *Nlrp1*$^{-/-}$ mice, from the Firmicutes and Bacteroidetes phyla (Fig. 2a). There was no difference in microbial community composition between WT and *Nlrp1*$^{-/-}$ mice, as determined by redundancy analysis (RDA), canonical correspondence analysis (CCA) and ADONIS using gender as covariate. Diversity and richness were not associated with genotype (multiple regression analysis, corrected for gender $p > 0.1$). To further establish which

of these may be associated with the DSS-colitis phenotype, we looked to see if they could confer protection to co-housed WT mice. To accomplish this, WT and *Nlrp1*$^{-/-}$ mice were co-housed for 4 weeks to allow the transfer of dysbiotic microbes, then subjected to DSS treatment. While single-housed WT mice displayed severe DSS-induced colitis as expected, WT mice that were co-housed with *Nlrp1*$^{-/-}$ mice and the single-house *Nlrp1*$^{-/-}$ mice showed reduced signs of colitis (Fig. 2b–d). Prior to DSS treatment, stool was collected from all mice for microbiome analysis using 16S ribosomal RNA sequencing. The microbial compositions of single-housed *Nlrp1*$^{-/-}$, co-housed WT and co-housed *Nlrp1*$^{-/-}$ mice were compared to single-housed WT mice. We hypothesized that as *Nlrp1*$^{-/-}$ mice possess a composition of gut bacteria that provide protection from DSS-induced colitis, the abundance of these bacteria should be increased in protected mice compared to singly housed WT. Interestingly, we found that all OTUs significantly upregulated in co-housed mice were from the order *Clostridiales*, including species from the *Lachnospiraceae* and *Ruminococcaceae* families (Fig. 2e). Similar species have been reported to generate high levels of butyrate[25], which are known to play a role in alleviating intestinal pathologies[26].

**Vancomycin or butyrate treatment of Nlrp1$^{-/-}$ mice**. To investigate whether bacteria of the *Clostridiales* order contribute to the protection of *Nlrp1*$^{-/-}$ mice against DSS-induced colitis, we treated WT and *Nlrp1*$^{-/-}$ mice with vancomycin (Gram-positive specific antibiotic) for 4 weeks prior to DSS administration. To ensure depletion of *Clostridiales* we performed quantitative

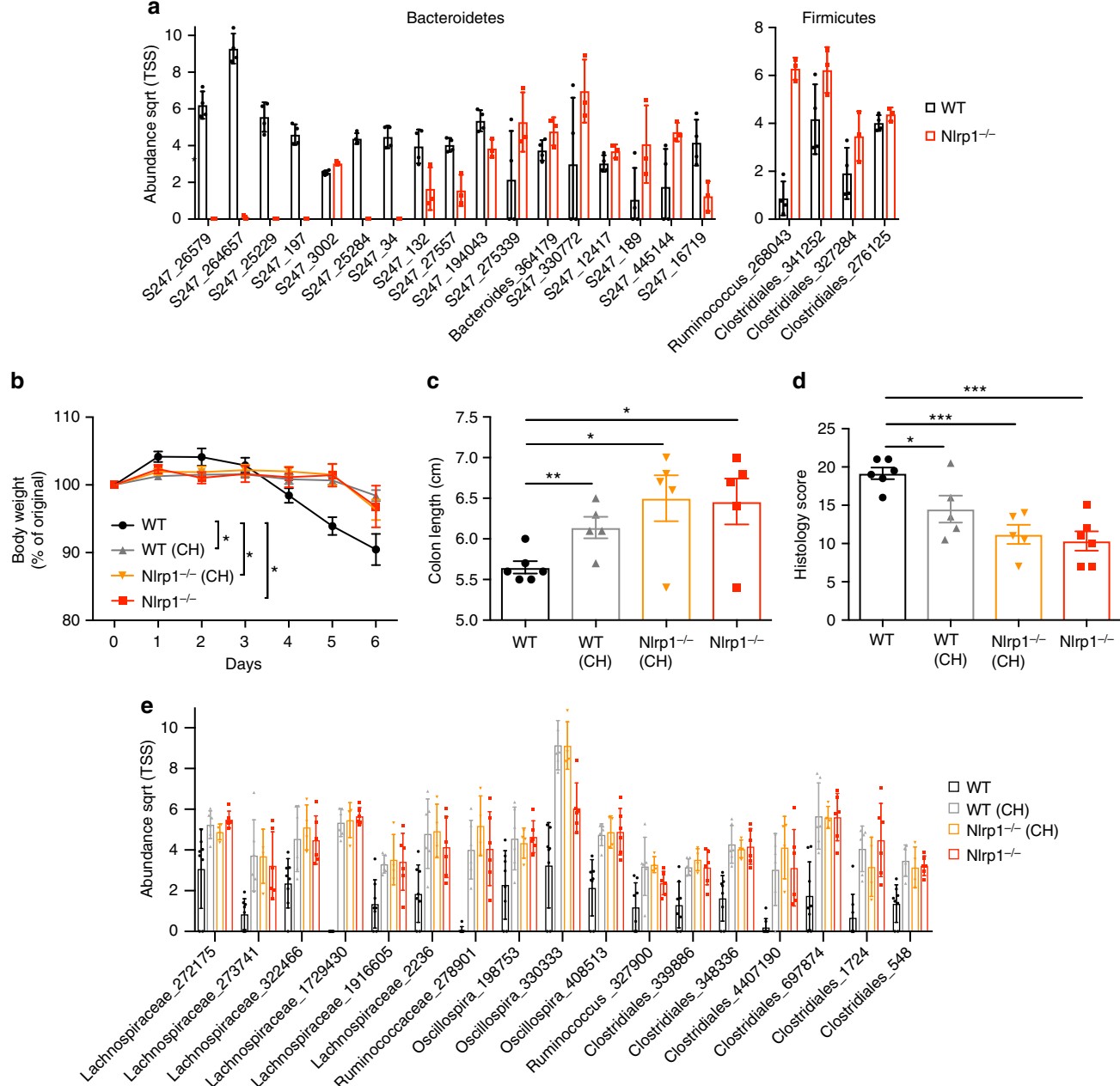

**Fig. 2** Microbiome dysbiosis in *Nlrp1*[−/−] mice is transferred to co-housed WT mice. **a** WT and *Nlrp1*[−/−] littermates were separated at weening for 6 weeks, then stool was collected and subjected to bacterial 16S analysis. Differentially abundant OTUs are presented. **b** WT mice were either single-housed or co-housed (CH) with *Nlrp1*[−/−] mice for 4 weeks prior to DSS treatment and weight loss, **c** colon length and **d** histology score were measured for signs of disease. **e** Prior to DSS treatment fecal samples were collected for bacterial 16S analysis. Comparison of 16S sequencing showing differentially abundant OTUs in 3 pair-wise comparisons made to WT mice. Data are representative of 2 independent experiments with 3–6 mice per group. Means ± SD (**a**, **e**), means ± SEM (**b–d**); *$p \leq 0.05$, **$p \leq 0.01$, ***$p \leq 0.001$. As determined by a two-tailed, unpaired *t*-test

real-time PCR (RT-PCR) on stool samples before and after vancomycin treatment with specific primers against one major group, *Clostridium coccoides*[27]. Treatment with vancomycin significantly reduced the abundance of coccoides in both WT and *Nlrp1*[−/−] mice (Fig. 3a). As expected, depleting the elevated bacteria from *Nlrp1*-deficient mice meant that they no longer had a protected DSS-colitis phenotype compared to WT mice (Fig. 3b, c). To determine whether the protective colitis phenotype of *Nlrp1*[−/−] mice was associated with increased butyrate production by the gut microbiota, the stools of WT and *Nlrp1*[−/−] mice, before and after vancomycin treatment, were solvent extracted using a cryomill and short chain fatty acids (SCFAs) analyzed by triple-quadruple

gas chromatography mass spectrometry. A significant increase of butyrate was observed in *Nlrp1*[−/−] stools compared to WT stools, which decreased in both cases upon vancomycin treatment (Fig. 3d). Significant differences in other SCFAs, such as propionate, were not observed in WT and *Nlrp1*[−/−] stools at steady state, although levels of propionate were reduced after vancomycin treatment (Fig. 3e). In addition, we did not observe any differences in the total number of CD45+ and CD4+ cells in the colon of vancomycin-treated WT and *Nlrp1*[−/−] mice, with the frequency of IFNγ+ (Th1), IFNγ+IL-17a+ (Th1/Th17) or IL-17a+ (Th17) cells being similar in both genotypes (Supplementary Fig. 3).

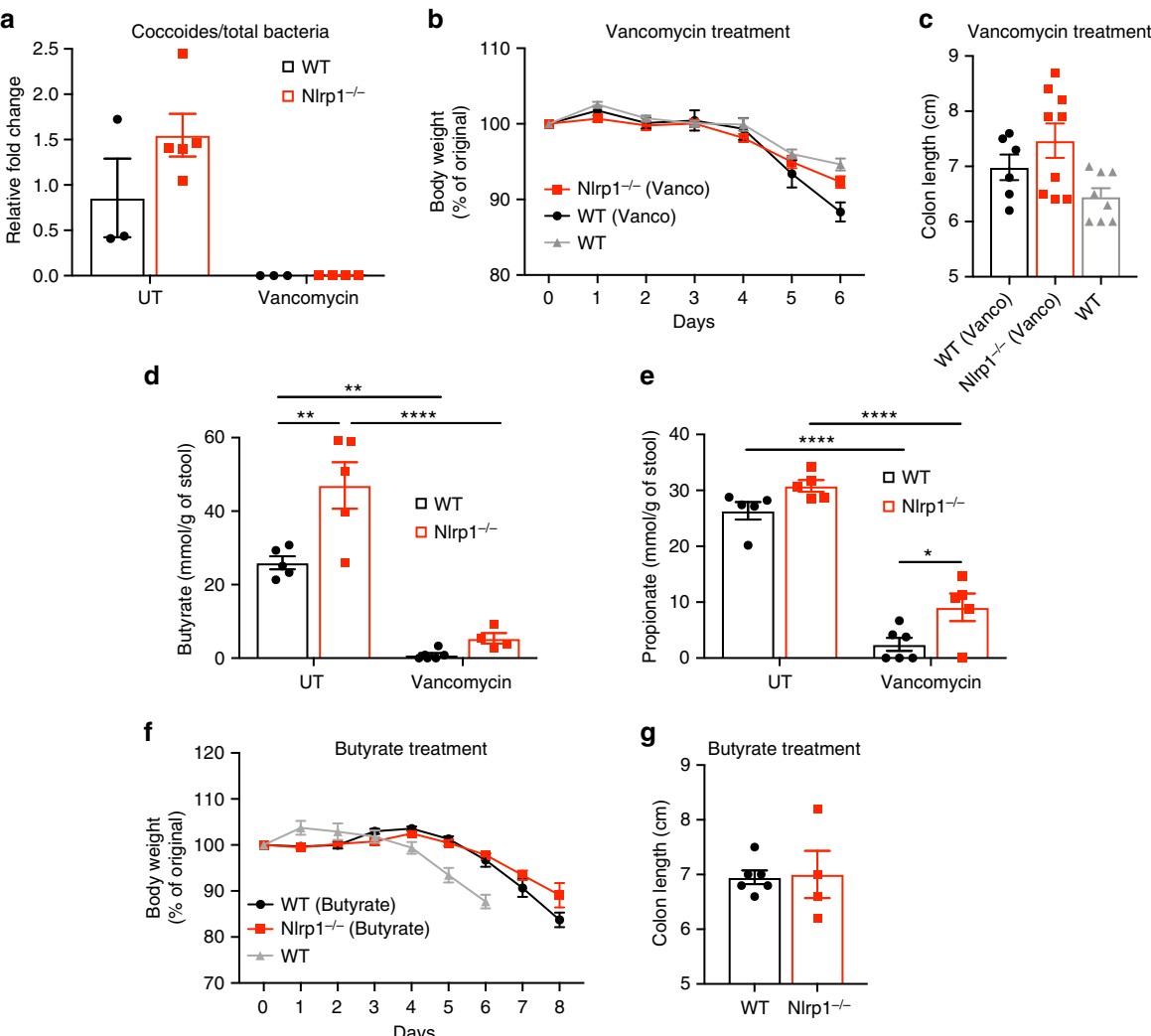

**Fig. 3** Vancomycin treatment or supplementation with butyrate ablates the Nlrp1 phenotype. **a** Stool from WT and $Nlrp1^{-/-}$ mice was harvested before and after vancomycin (50 mg/L) treatment for 4 weeks. Bacterial DNA from stool was isolated from untreated (UT) and vancomycin-treated mice, and depletion of the Coccoides group (belonging to the *Clostridiales* phylum) was confirmed using specific 16S primers by quantitative PCR. Results were normalized to the total bacteria present in the stool. **b** Vancomycin-treated mice were subjected to 3% (w/v) DSS for 6 days and disease severity was measured according to percentage weight loss and **c** colon length. Short chain fatty acid analysis was performed on fecal samples collected from WT and $Nlrp1^{-/-}$ mice before (UT) and after vancomycin treatment. The concentration of **d** butyrate and **e** propionate was measured by gas chromatography mass spectrometry. **f** WT and $Nlrp1^{-/-}$ mice were supplemented with 2% butyrate in drinking water ad libitum for 28 days, and then subjected to 2.5% (w/v) DSS for 6 days and disease severity measured according to percentage weight loss or **g** colon length. Data are representative of 3 independent experiments with 3–6 mice per group. Means ± SEM; *$p \leq 0.05$, **$p \leq 0.01$, ****$p \leq 0.0001$. A two-tailed unpaired *t*-test was used to determine statistical significance between two groups; and a two-way ANOVA with Tukey's post-hoc comparisons was performed on data that involved more than two comparisons

To provide evidence that butyrate is the causal link between elevated levels of *Clostridiales* and the protection of $Nlrp1^{-/-}$ mice against DSS-induced colitis, we performed butyrate supplementation by feeding mice 2% butyrate in drinking water for 4 weeks before initiating DSS treatment. This regime largely protected the WT mice from DSS-induced colitis, and as expected, there was no longer a difference to $Nlrp1^{-/-}$ mice also supplemented with butyrate, with regards to weight loss and inflammation in the colon (Fig. 3f, g). Together, these results indicate that increased colonization by bacteria from the order *Clostridiales* lead to increased levels of butyrate which protect $Nlrp1^{-/-}$ mice from DSS-induced colitis.

**Nlrp1 exacerbates DSS-colitis independent of IL-1R signaling.** To confirm the pathogenic role of NLRP1 activation during DSS-induced colitis, we utilized a mouse model that displays

hyper-activated Nlrp1a. We hypothesized that $Nlrp1a^{Q593P/Q593P}$ mice would be highly susceptible to DSS-induced colitis. Further, we genetically manipulated downstream signaling components such as IL-1 or IL-18 to examine which pathway contributed to NLRP1-mediated disease during IBD. Mice with hyper-activation of Nlrp1a, but lacking the IL-1 receptor ($Il-1r^{-/-}Nlrp1a^{Q593P/Q593P}$), were highly susceptible to DSS-induced colitis compared to littermate controls ($Il-1r^{-/-}Nlrp1a^{Q593P/+}$), and had to be ethically killed by day 5 (Fig. 4a). Additionally, $Il-1r^{-/-}Nlrp1a^{Q593P/Q593P}$ had a significant reduction in colon length and increased mucosal damage and inflammatory cell infiltrate highlighted by an increased histology score (Fig. 4b–d). Given the increased abundance of *Clostridiales* in $Nlrp1^{-/-}$ mice, we expect that these bacteria would be deficient in mice with an activating mutation in the gene. In agreement with this, it was not possible to transfer the phenotype of $Il-1r^{-/-}Nlrp1a^{Q593P/Q593P}$

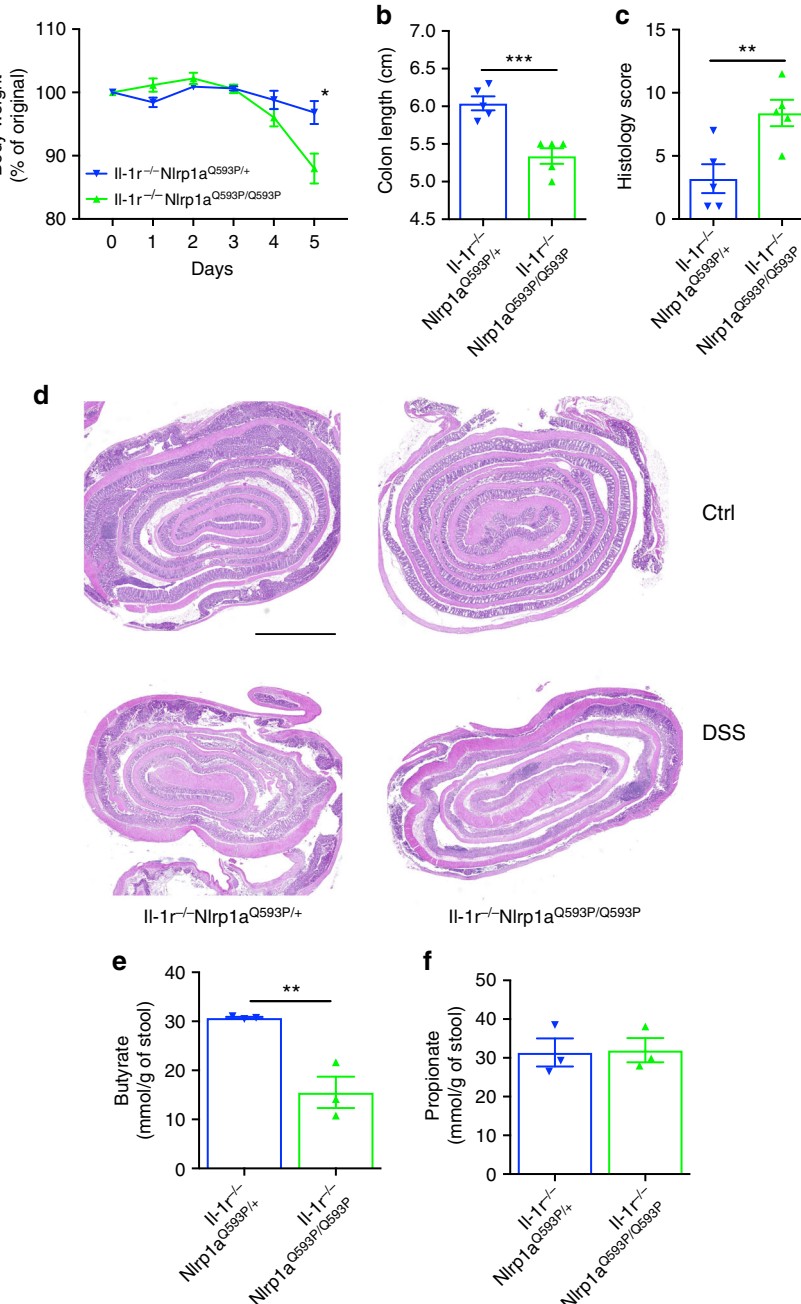

**Fig. 4** Hyper-activation of Nlrp1 exacerbates DSS-induced colitis independent of IL-1R signaling. *Il-1r*$^{-/-}$*Nlrp1a*$^{Q593P/+}$ and *Il-1r*$^{-/-}$*Nlrp1a*$^{Q593P/Q593P}$ mice were given 3% (w/v) DSS for 6 days. **a** The percentage weight loss, **b** colon length and **c** histology score for epithelial damage and inflammatory cell infiltrate of H&E-stained sections of the colon were measured for disease severity. **d** Representative H&E-stained sections of the distal colon from *Il-1r*$^{-/-}$*Nlrp1a*$^{Q593P/+}$ and *Il-1r*$^{-/-}$*Nlrp1a*$^{Q593P/Q593P}$ mice, scale bar = 2 mm. Short chain fatty acid analysis was performed on fecal samples collected from WT and *Nlrp1*$^{-/-}$ mice before (UT) and after vancomycin treatment. The concentrations of **e** butyrate and **f** propionate were measured by gas chromatography mass spectrometry. Data are representative of 2 independent experiments with 3–5 mice per group. Means ± SEM; *$p \leq 0.05$, **$p \leq 0.01$, ***$p \leq 0.001$. As determined by a two-tailed, unpaired *t*-test

mice by co-housing, suggesting that the bacteria were not present in their stool. This was supported by analysis of SCFA levels in the stool of control and *Il-1r*$^{-/-}$*Nlrp1a*$^{Q593P/Q593P}$ mice, which showed that in line with increased colitis severity, butyrate was significantly reduced in *Il-1r*$^{-/-}$*Nlrp1a*$^{Q593P/Q593P}$ stool, while propionate was not altered (Fig. 4e, f). This indicates that Nlrp1 activation contributes to DSS-induced colitis, with decreased butyrate production, even in the absence of IL-1 signaling.

FACS analysis further highlighted that *Il-1r*$^{-/-}$*Nlrp1a*$^{Q593P/Q593P}$ mice have an increase in the total number of CD4+ T cells present

in the cLP, during both SS and DSS-induced colitis (Supplementary Fig. 4a). Interestingly, at SS *Il-1r*$^{-/-}$*Nlrp1a*$^{Q593P/Q593P}$ mice displayed a significant increase in the frequency of IFNγ-producing CD4+ T cells, which doubled upon DSS treatment in the cLP (Supplementary Fig. 4b-c). However, no significant differences were observed in the frequencies of IFNγ+IL-17A+ and IL-17A+CD4+ T cells in the cLP and spleen at SS or during DSS. The increase in the IFNγ-producing CD4+ T cells was associated with a significant increase in IL-18 production from cells present in the cLP (Supplementary Fig. 4d), with IL-18 already

reported to be a potent regulator of IFNγ-producing CD4 +T cells[28]. Since there is a role for NLRP1 in the non-hematopoietic compartment, we also measured IL-18 in the epithelial fraction; however, due to severe epithelial damage, the recovery of viable epithelial cells was low, thus accounting for low IL-18 production measured by enzyme-linked immunosorbent assay (ELISA) (Supplementary Fig. 4d). These data suggest that activation of Nlrp1 mediates DSS-induced colitis, possibly through increased IL-18 production, which is associated with the expansion of IFNγ-producing CD4+ T cells in the colon.

**Deleting IL-18 reduces Th1 response from Nlrp1 in DSS-colitis.** Although genetic deletion of IL-1R did not rule out a contribution from IL-1β downstream of Nlrp1 in DSS-colitis, it did implicate IL-18 in this process. Unfortunately, $Il$-$18^{-/-}Nlrp1a^{Q593P/Q593P}$ mice have a spontaneous inflammatory phenotype and die prematurely at approximately 7 weeks of age[13]. For this reason we are unable to examine their phenotype in the mouse model of DSS-induced colitis, and instead we genetically deleted the gene encoding IL-18 on the $Il$-$1r^{-/-}Nlrp1a^{Q593P/Q593P}$ background. In the absence of IL-18 signaling, the activation of Nlrp1a did not lead to an increase in disease pathology, indicated by no significant changes in colon length or histopathological scores (Fig. 5a–d). While we observed a modest increase in the absolute number of CD4+ T cells in the cLP of $Il$-$1r^{-/-}Il$-$18^{-/-}Nlrp1a^{Q593P/Q593P}$ mice during DSS (Supplementary Fig. 5a), the frequencies of IFNγ+, IFNγ+IL-17A+ and IL-17A+ CD4+ T cells at SS and during DSS was not significantly different in the cLP and spleen (Supplementary Fig. 5b-c). This shows that IL-18 signaling after Nlrp1 activation

increases the disease severity of DSS-colitis, associated with increased IFNγ-producing Th1 cells.

**Increased NLRP1 in UC correlates with decreased *Clostridiales*.** To address the relevance of NLRP1 in human IBD, we performed microarray analysis on inflamed human colon biopsies from patients with UC and CD compared to healthy controls. We observed a significant fold increase in *NLRP1* gene expression in the inflamed regions of the sigmoid colon and rectum of UC patients compared to healthy controls, which was not observed in CD patients (Fig. 6a, b). Although we do not expect NLRP1 expression to affect *IL-18* at the RNA level, there was a positive correlation to *IFNγ* gene expression in the inflamed distal regions of the colon (Fig. 6c). The 16S microbial sequencing was performed on rectal mucosal biopsies from healthy control and UC patients, and then tested for a negative correlation to *NLRP1, IL-18* or *IFNγ* expression. This analysis revealed that of the eight OTUs that were statistically negatively correlated to *NLRP1* expression (Supplementary Table 1), half were of the order *Clostridiales* (Supplementary Table 1), and the most significant of which, *Faecalibacterium prausnitzii*, is documented to have anti-inflammatory effects in models of IBD[29]. *F. prausnitzii* and other *Clostridiales* also dominate the list of OTUs that are negatively correlated with *IL-18* and *IFNγ* expression (Supplementary Tables 2 and 3). Taken together, these human data confirm our observations from mice that NLRP1 may be involved in promoting a Th1 response in the colon to deplete beneficial butyrate-producing commensals and facilitate the pathogenesis of UC.

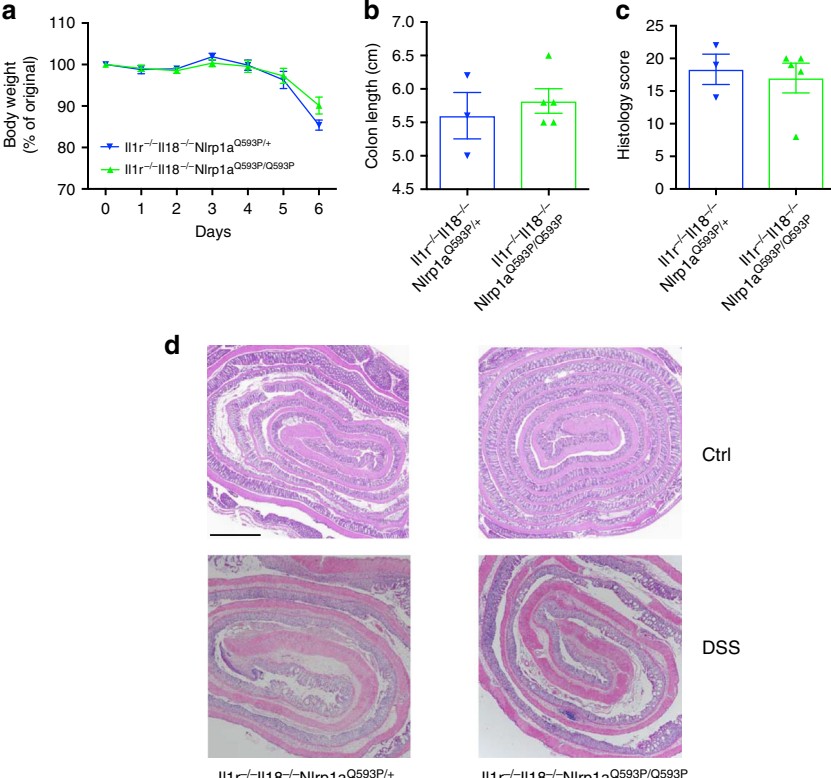

**Fig. 5** Deletion of IL-18 removes the hyper-activated Nlrp1 DSS-induced colitis phenotype. $Il$-$1r^{-/-}Il$-$18^{-/-}Nlrp1a^{Q593P/+}$ and $Il$-$1r^{-/-}Il$-$18^{-/-}Nlrp1a^{Q593P/Q593P}$ mice were given 3% (w/v) DSS for 6 days. **a** The percentage weight loss, **b** colon length and **c** histology score for epithelial damage and inflammatory cell infiltrate of H&E-stained sections from the colon were measured for disease severity. **d** Representative H&E-stained sections of the distal colon from $Il$-$1r^{-/-}Il$-$18^{-/-}Nlrp1a^{Q593P/+}$ and $Il$-$1r^{-/-}Il$-$18^{-/-}Nlrp1a^{Q593P/Q593P}$ mice, scale bar = 1 mm. Data are representative of 2 independent experiments with 3–5 mice per group. Means ± SEM

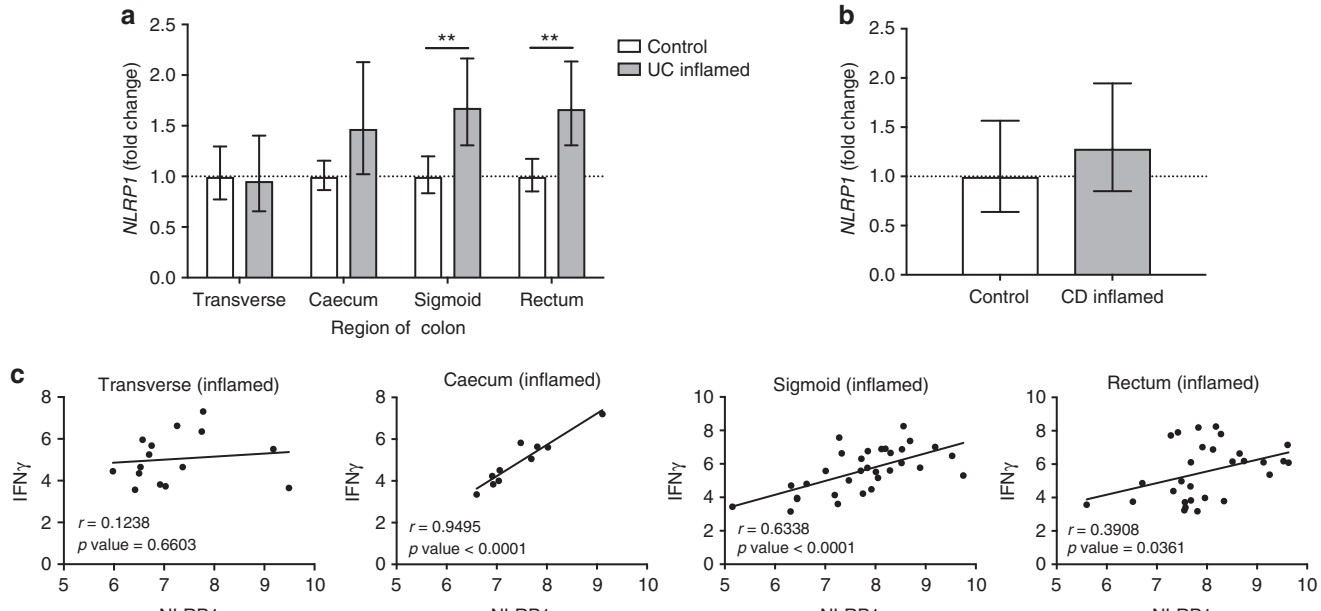

**Fig. 6** *NLRP1* gene expression is augmented in human ulcerative colitis which positively correlates with *IFNγ* gene expression in inflamed regions of the colon. Microarray analyses were performed on colon biopsies from healthy controls ($n = 22$), and patients with ulcerative colitis or Crohn's disease. Gene expression data expressed as relative fold change compared to healthy controls from **a** transverse ($n = 15$), cecum ($n = 10$), sigmoid ($n = 32$) and rectum ($n = 29$) regions of the colon from patients with ulcerative colitis and **b** ileum ($n = 25$) from patients with Crohn's disease. Correlation between *NLRP1* and *IFNγ* gene expression in various regions of the colon of **c** inflamed regions from ulcerative colitis patients. Means ± 95% asymmetric confidence interval; \*\*$p \leq 0.01$. A one-way ANOVA and Tukey's post-hoc comparisons were performed on data that involved more than two comparisons, while a two-tailed, unpaired *t*-test was performed on data containing two comparisons. Pearson's correlation coefficient was used to generate the *r* value and the *p* values denoting the significance were estimated from a paired *t*-test

## Discussion

We have shown that loss of the Nlrp1 inflammasome ameliorates DSS-induced colitis by promoting expansion of beneficial gut microbes belonging to the *Clostridiales* phylum, with concomitant increased butyrate production in the colon. Our results contrast with those of Williams et al.[30] in a recent publication concerning mice that are genetically deficient for *Nlrp1b*. These *Nlrp1b*$^{-/-}$ mice displayed exacerbated DSS-induced colitis which was mediated by reduced levels of both IL-1β and IL-18. One explanation for this observation could be related to the genetics of mouse strains used. Our *Nlrp1*$^{-/-}$ mice lack all three alleles of *Nlrp1*, while the *Nlrp1b*$^{-/-}$ mice still encode functional *Nlrp1a*, although it is from the parental 129 strain, where it is not expressed in macrophages and the jejunum[14,31]. To clarify this, we studied mice where only *Nlrp1a* was deleted, and again these mice were protected from DSS-colitis. Our *Nlrp1a*$^{-/-}$ mice were originally made on the 129 background and then backcrossed to C57BL/6 and hence the *Nlrp1b* allele is still active, with respect to activation by LT (Lethal Toxin). Moreover, these experiments were performed in a separate animal facility, indicating that the protection to DSS-colitis afforded by loss of *Nlrp1a* is not restricted to a single set of environmental conditions. To independently confirm these results, we have also shown that hyper-activation of the *Nlrp1a* allele specifically exacerbates the pathogenesis of DSS-induced colitis in C57BL/6 mice. In summary, our results support a dominant role for *Nlrp1a* in this mouse model of IBD.

We have shown that IL-18 is involved in the exacerbation of DSS-colitis after activation of NLRP1. The role of IL-18 in IBD has recently been clarified by studies which showed the importance of cell specificity for IL-18 signaling in colitis, as deletion of IL-18 from epithelial cells, not myeloid cells, conferred protection against DSS-induced colitis[9]. This also agrees with our

observation that deletion of NLRP1 from non-hematopoietic cells conferred protection against DSS-induced colitis. In our studies, NLRP1 activity and IL-18 production in colitic mice was associated with increased IFNγ-producing CD4+ T cells in the cLP. To support this notion, our human data confirm that *NLRP1* gene expression is significantly elevated in inflamed distal regions of the colon from patients with UC, and that there is a positive correlation between *NLRP1* and *IFNγ* gene expression. Together, these results are consistent with the observation that there is an increase in IL-18 and Th1 responses in human IBD[4,32].

Our understanding for the roles of ASC and Caspase-1 during DSS-induced colitis are currently being re-defined. This was necessary because earlier studies did not adequately normalize the microbiota of these strains, for example by using littermate control mice, or the F1/F2 progeny thereof, as we have done in this study. These new data suggest that in fact, mice deficient for ASC containing inflammasomes bear no impact on the outcome of DSS-colitis[11]. That result is consistent with our study, as NLRP1 can function independently of ASC in mice. Furthermore, a new Caspase-1$^{-/-}$ allele, where Caspase-11 is still functional, was examined in an enhanced barrier facility, and found to be protected against DSS-colitis[12]. This is broadly consistent with our observations, with the exception that no difference in *Clostridales* was seen in that study. One potential explanation for this is that Caspase-1 integrates signals from a wide variety of inflammasome sensors, for example NLRC4, which may have a dominant or confounding effect.

Dysbiosis in the gut microbiota of IBD patients is commonly observed and is often associated with reduced colonization by bacteria from the *Clostridium* cluster *XIVa* and *IV*[33], which includes bacteria from the *Lachnospiraceae* and *Ruminococcaceae* families that were abundant in *Nlrp1*$^{-/-}$ mice. These bacteria have also been reported to be good producers of butyrate[25], which

is consistent with increased butyrate production in fecal samples isolated from $Nlrp1^{-/-}$ mice. Additionally, we have also demonstrated that depletion of *Clostridiales* using the antibiotic vancomycin reduces butyrate levels and increases DSS-induced disease severity, such that there is no longer a difference between WT and $Nlrp1^{-/-}$ mice. Moreover, there was a significant negative correlation between *NLRP1, IL-18* or *IFNγ* expression and members of the *Clostridiales* order in human intestinal biopsies. The species that was most significantly correlated to *NLRP1* expression and is also correlated to *IL-18* expression (*F. prausnitzii*) is a known butyrate-producing commensal that is anti-inflammatory in models of IBD and is decreased in UC compared to healthy controls[29]. To our knowledge, this is the first description of NLRP1 regulating microbial species from the *Clostridiales* order that are beneficial for IBD. This is a unique feature for any pattern recognition receptor, which are usually described as a limiting factor for the colonization of pathogenic strains. Our results suggest that it may be possible to therapeutically target the NLRP1 inflammasome pathway to increase butyrate-producing commensals in the gut of patients who are otherwise deficient, and thus prevent IBD.

## Methods

**Human colon biopsy collection.** Human intestinal pinch biopsies were collected by a single operator (G.L.R.-S.) at the time of colonoscopy using a standard biopsy forcep technique (Boston Scientific Radial Jaw 4 (2.8 mm)). All subjects gave written informed consent, and the study was approved by the Human Research Ethics Committees of the Royal Brisbane and Women's Hospital, Brisbane, Australia (HREC/14/QRBW/323), and the QIMR Berghofer Medical Research Institute, Brisbane, Australia (HREC/P692). Patients included in the study were all under the management of the Inflammatory Bowel Disease team with an established diagnosis and well-characterized disease course. Healthy controls were recruited from gastroenterology clinics and only included individuals who were undergoing a screening colonoscopy for a family history of colorectal cancer, and in whom the procedure was normal. A total of 101 subjects (33 CD, 56 UC and 22 healthy controls) were included in the study, the characteristics of which are included as Supplementary Table 4. Biopsies were collected from representative involved and adjacent uninvolved intestinal segments, including the terminal ileum (CD), the cecum, transverse colon, sigmoid colon and rectum (UC). Biopsies were snap frozen immediately and stored at −80 °C for RNA and DNA extraction. Adjacent biopsies were also taken from these segments for histological analysis. An inflammation score was generated for each biopsy site in each case, based upon a validated scoring system[34].

**Mice.** Mice that lack all three $Nlrp1$ genes in the murine $Nlrp1abc$ locus (C57BL/6 background) and $Il-1r^{-/-}Nlrp1a^{Q593P/Q593P}$ mice have been previously described[13], and were housed and bred in the same animal facility (WEHI). Mice that lack $Nlrp1a^{-/-}$ were generated by replacing exon 3 with a neomycin selection cassette on the 129 background and then backcrossed to C57BL/6 (C57BL/NTac background). These mice were housed and bred in a separate facility (Yale). All animal experiments were performed under the standards of, and were approved by, the Walter and Eliza Hall Institute Animal Ethics Committee, or the Yale University Institutional Animal Care and Use Committee.

**Bone marrow chimeras.** C57BL/6 WT mice or $Nlrp1^{-/-}$ mice at 6 weeks of age were reconstituted with $5 \times 10^6$ WT mice or $Nlrp1^{-/-}$ bone marrow cells. Recipient mice received two 5.5 Gy doses of irradiation given 3 h apart and were treated with Neomycin for 3 weeks. Mice were treated with DSS 12 weeks after irradiation.

**Induction of DSS-induced colitis.** To induce acute colitis in mice, 3% (w/v) DSS (molecular mass 40-50 kDa; Affymetrix) was dissolved in sterile water and provided to the 8–12-week-old mice ad libitum twice for 3 days each (for 6 days in total) followed by normal drinking water until day 8 or when mice had lost 20% of their initial body weight. The disease severity parameters for this study were based on the percentage weight loss, changes in colon length and histopathology scores of the colon. Some mice were treated with 50 mg/L Vancomycin (Sigma) in drinking water for 4 weeks to deplete intestinal microbiota prior to DSS challenge. For butyrate supplementation, 2% (w/v) sodium butyrate (Sigma) was administered in drinking water for 4 weeks, prior to, and then continued throughout, DSS treatment. All animal experiments were age- and sex-matched appropriately.

**Histopathology and immunohistochemistry.** Disease severity was determined by histopathology scoring of a hematoxylin and eosin (H&E)-stained colon in a blinded fashion, which is characterized by the recruitment of inflammatory cell

infiltration (score 0–3) and epithelial damage in the colon (score 0–5). The presence of occasional inflammatory cells in the lamina propria was scored as 0, increased numbers of inflammatory cells in the lamina propria was scored as 1, infiltration of inflammatory cells into the submucosa was scored as 2 and transmural extension of the infiltrate was scored as 3. No epithelial damage was scored as 0, hyperproliferation of the mucosa was scored as 1, less than 50% crypt loss was scored as 2, more than 50% crypt loss was scored as 3, 100% crypt loss was scored as 4 and the presence of an ulcer was scored as 5.

**Organ cultures and cytokine measurements.** The distal colon (2 cm from the rectum) was isolated and flushed with sterile phosphate-buffered saline (PBS). Epithelial cells were isolated using dissociation media (Hanks free+5 mM EDTA) at 37 °C with gentle agitation for 30 min and cultured separately for 6 h. The supernatants were analyzed for IL-18 by ELISA using IL-18-specific antibodies (R&D Systems, MN, USA). The concentration of IL-18 from the cEC (colonic epithelium) and cLP were normalized to the total amount of protein present in cEC measured by BCA protein assay (Life Technologies, CA, USA) and the weight of tissue collected, respectively.

**Isolation of colonic lamina propria.** Whole colons were isolated and cut longitudinally from the distal colon to the cecum and thoroughly washed in sterile PBS. The epithelial layer was isolated in dissociation media (Hanks free+5 mM EDTA) at 37 °C with gentle agitation for 30 min. To isolate lymphocytes from the cLP, the remaining colon was digested in digestion media (RPMI, 2% fetal calf serum, 1 mg/mL Collagenase III (Worthington Biochemical Corporation, NJ, USA), 0.4 units Dispase (Life Technologies, CA, USA), 1 μg/mL DNase (Life Technologies, CA, USA)) at 37 °C with gentle agitation for 1 h. Lymphocytes were isolated by performing a 40%/80% Percoll gradient followed by centrifugation at $900 \times g$ for 20 min with no brakes. Lymphocytes are found at the 40/80% interface.

**FACS analysis of T-cell subsets.** Lymphocytes were isolated and re-stimulated with 50 μg/mL phorbol myristate acetate + 1 μg/mL ionomycin and Brefeldin A at 37 °C and 5% $CO_2$ for 4 h. Cells were washed and incubated with anti-Fc receptor and stained for FACS analysis using the Foxp3/transcription factor staining buffer set (eBioscience, CA, USA) with a combination of antibodies: CD45-APC (1:400 clone A20.1), CD3-A700 (1:200 clone KT3-1.1) and CD4-FITC (1:400 clone GK1.5) obtained from our monoclonal antibody facility; IFNγ-PercpCy5.5 (1:300 clone XMG1.2) and IL-17A-pacific blue (1:300 clone 17B7) purchased from eBioscience; and Live/Dead fixable dead cell stain (Life Technologies, CA, USA).

**RNA isolation and microarray of human intestinal biopsies.** Total RNA was isolated from 2–3 pinch biopsy samples per site using Qiagen RNeasy kits (QIAGEN, Valencia, CA) as per the manufacturer's instructions. RNA integrity was determined using Experion™ Automated Electrophoresis System (Bio-Rad Laboratories, Hercules, CA). RNA quality indicator (RQI) number, range and median, for each disease group, was as follows: CD (range, 6.3–9.5; median, 7.7), UC (range, 4.4–9.7; median, 8.1) and healthy controls (range, 6.7–9.2; median, 8.3).

Using a concentration of 50 ng/μL, 1 μL was used for the Ovation RNA Amplification System v2 (Nugen). Poly-A control spike-ins (Affymetrix) were added prior to reverse transcription followed by SPIA (Single Primer Isothermal Amplification). The reaction was split equally prior to SPIA amplification which was performed in two stages: 30 min at 48 °C followed by addition of 3 μL WB reagent before a further 30 min at 48 °C. The complementary DNA (cDNA) product was purified (Qiagen QIAquick Reaction Cleanup Kit) and quantified by spectroscopy (Nanodrop 2000). Only samples yielding >5 μg were further processed using the Encore Biotin Module (Nugen) where 4.4 μg of cDNA was fragmented and biotinylated. This was added to Oligo B2, 20× Eukaryotic Hybridisation Controls, Herring Sperm DNA, acetylated bovine serum albumin, 2× Hybridisation buffer, 100% dimethyl sulfoxide and nuclease-free water as described for use with AffymetrixGeneChip® U133 Plus 2.0 arrays by Affymetrix. Then, 200 μL of the hybridisation cocktail was incubated with the array 18 h ± 2 h at 45 °C, 60 rpm rotation in a 645 Hybridization oven (Affymetrix). After incubation the hybridization solution was recovered and the array filled with Wash Buffer A (6× SSPE, 0.01% Tween 20) prior to wash/stain/scan on a Fluidics Station 450 using the EukGE-WS2v5_450 script, with all reagents prepared according to Affymetrix guidelines. Arrays were scanned on a GeneChip Scanner 3000–7G.

**16S metagenomics and bioinformatics analysis of mouse stool.** Bacterial DNA was isolated from mouse fecal samples using the QIAamp DNA Stool Mini Kit (Qiagen, Limberg, The Netherlands). RT-PCR was performed to quantify coccoides using specific primers (forward (Fwd) AAATGACGGTACCTGACTAA and reverse (Rev) CTTTGAGTTTCATTCTTGCGAA) compared to total bacteria (Fwd GGTGAATACGTTCCCGG and Rev TACGGCTACCTTGTTACGACTT). The V3–V4 regions of the 16S ribosomal RNA gene were amplified using the primer pair 341F-805R (Fwd GTGACCTATGAACTCAGGAGTCCCTACGGGNGGC WGCAG and Rev CTGAGACTTGCACATCGCAGCGACTACHVGGGTA TCTAATCC) under PCR conditions: 95 °C for 10 min, 18 cycles (95 °C for 30 s, 58 °C for 30 s and 72 °C for 20 s) 72 °C for 7 min. Amplicons were purified and libraries were prepared by annealing Illumina index primers using PCR conditions: 95 °C for 2

min, 24 cycles (95 °C for 15 s, 60 °C for 30 s and 72 °C for 30 s) and 72 °C for 7 min. Libraries were sequenced on the MiSeq using 2 × 310 nucleotide long reads.

Paired-end reads were joined using pear (v0.9.6) and primer sequences were removed using cutadapt (v1.15). Reads were imported into QIIME (v1.9.1) and clustered into OTU (pick_open_reference_otus.py, Greengenes v13.8, 97% identity threshold). Representative sequences were assigned to taxonomy with uclust. The taxonomic profile was imported into Calypso v8.68 using total sum normalization and cumulative-sum scaling (CSS). RDA, CCA and ADONIS were used to elucidate relationship between the overall microbial composition (OTU level) and the explanatory variables genotype and gender. A generalized linear regression model (GLM) including gender as factor was applied to identify significantly different OTUs. Shannon diversity and richness indices were calculated at OTU level and used as dependent variables in a GLM to assess associations between diversity and genotype. For the co-housing experiment t-test was applied. All p values were corrected using Benjamini Hochberg. Principal coordinate analysis was performed using the Bray–Curtis distance metric.

**16S metagenomics analysis of human intestinal biopsies**. Bacterial DNA was isolated from rectal mucosal biopsies using QIAGEN DNeasy Blood & Tissue kit (Hilden, Germany). DNA samples were processed by the Australian Genome Research Facility for microbial diversity profiling using the Illumina MiSeq platform, with 16S sequences amplified using primer pair 341F-806R. Paired-end Illumina reads were merged with the software PEAR v0.9.6[35]. Subsequently, the merged reads were processed using the QIIME pipeline v1.91[36]. Sequences were filtered for quality using default setting with a quality phred threshold of at least Q20 and at least 75% of original length. Forward and reverse PCR primers were removed allowing one mismatch. Sequences were clustered into OTUs using an identity threshold of 97% and the Greengenes database v13.5 as reference. Subsequently, unclustered sequences were grouped using QIIME de novo approach. Finally, representative sequences of each OTU were assigned to a taxonomic lineage using uclust consensus taxonomy assigner[37] and the Greengenes database v13.5. Representative sequences were searched for matches to sequences assigned to Eukaryotes in the National Center for Biotechnology Information nucleotide (NCBI nt) database using blastn[38] and excluded from the analysis. OTUs with a size of at least 10 sequences were used for the analysis. The taxonomic profile was imported to Calypso[39] using the default settings with CSS normalization and log transformation. The normalized table was used in the software R to calculate Pearson's correlation between the normalized expression of NLRP1, IL-18 or IFNγ and the predicted OTUs.

**Short chain fatty acid analysis**. Fecal pellets (10–20 mg) were weighed in a cryomill tube (FastPrep-24, MP Biomedicals), and then suspended in 0.5% orthophosphoric acid in water (250 µL, v/v) containing the internal standard 4-methylvaleric acid (100 µM) to account for any losses during the extraction procedure. The extraction was further supplemented with $^{13}C$-labeled SCFA ($^{13}C_2$-acetate (50 µM), $^{13}C_3$-propionate (10 µM) and $^{13}C_3$-butyrate (10 µM)) to allow absolute quantitation of these SCFAs. The pellet was homogenized using a cryomill (Precellys 24/Cryolys, Bertin Technologies) at 6800 rpm for 30 s with 3 cycles with 45 s intervals at 0 °C. Samples were centrifuged at 14,000 rpm for 10 min at 4 °C and 200 µL of the supernatant mixed with equal volumes of ethyl acetate, prior to analysis on an Agilent 7890 GC-QQQ-MS (Agilent Technologies, Australia).

## Data availability

The datasets generated during and/or analyzed during the current study are available from the corresponding author on reasonable request.

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

## Acknowledgements

We thank L. Cengia and R. Lane for excellent technical assistance, and L. Scott, C. Hay and R. Crawley for outstanding animal husbandry. This work was supported by: Australian National Health and Medical Research Council (NHMRC) Project Grants (1057815, 1099262), Program Grants (461219, 1054618), fellowships (to B.T.K., S.L.M., M.J.M.) and an Independent Research Institutes Infrastructure Support Scheme Grant (361646). Fellowships from the Victorian Endowment for Science Knowledge and Innovation (to S.L.M.), HHMI-Wellcome International Research Scholarship (to S.L.M.), the Australian Research Council (to B.T.K.), the Sylvia and Charles Viertel Foundation (to B.T.K. and S.L.M.), the WEHI Centenary Fellowship (to C.-H.Y.) and Ormond College's Thwaites Gutch Fellowship in Physiology (to C.-H.Y.), the WEHI Centenary Dyson Bequest (to T.L.P.), and the Victorian Cancer Agency (to T.L.P.); the Australian Cancer Research Fund, the Australian Phenomics Network, the Ian Potter Centre for Genomics and Personalized Medicine and a Victorian State Government Operational Infrastructure Support Grant. The human microarray work was supported by a research grant from *Amgen* (South San Francisco, CA 94080). S.L.M. receives funding from Glaxosmithkline.

## Author contributions

H.T., C.-H.Y., L.A.S., M.R.d.Z., M.L.K., C.R.H., F.S.-F.-G., A.P., L.A.M., S.A.W., Y.O., S.C. C., K.K., K.A.K., D.P.D.S., M.G. and S.L.M. performed or assisted with experimentation. H.T., C.H.-Y., J.S.P., M.Z., L.F.W., M.J.M., R.A.F., M.G., B.T.K., A.T.P., T.L.P., G.L.R.-S.

and S.L.M. were involved in experimental analysis and interpretation. All authors contributed to the writing of this manuscript.

## Additional information

**Competing interests:** The authors declare no competing interests.

