## [Peer Review File · Nature Communications]

Reviewers' comments:

Reviewer #1 (PRR, inflammasome, bacterial infection)(Remarks to the Author):

This manuscript by Tye and colleagues investigates the role of NLRP1 in regulating intestinal homeostasis in a model of DSS colitis as well as in a model of colitis-associated cancer. The authors find that NLRP1 exacerbates pathology in terms of weight loss and inflammation in these settings, and find using 16S ribosomal profiling, that the *Nlrp1*^{-/-} genetic background is associated with an expansion of Clostridiales and Lachnospiraceae, whose presence has been linked to protection from inflammation in other settings. The authors further demonstrate that absence of NLRP1 is associated with increased levels of butyrate in the colon, and that supplementation with butyrate limits the DSS phenotype of the wild-type mice. The authors also provide evidence that NLRP1 exacerbates carcinogenesis in an AOM/DSS model of chronic inflammation, as *Nlrp1* deficiency is associated with reduced frequency and size of polyps forming in the colon following AOM/DSS treatment.

Interestingly, the authors report, using genetic analyses in which they analyze the effects of an activating NLRP1 mutation on colonic inflammation in a DSS model, that the pathological effect of NLRP1 depends on IL-18, rather than IL-1. Finally, the authors find increased expression of NLRP1 in particular regions of the colon in human ulcerative colitis patient samples, and observe a positive correlation in expression levels of NLRP1 with IFN γ these biopsies.

The findings in this manuscript are interesting, and the observation that NLRP1 contributes to intestinal pathology in several settings is novel. In several instances key controls are needed to strengthen and clarify the observations. Butyrate production by Clostridia has been previously demonstrated, but interestingly there appears to be some debate in the literature as to whether butyrate supplementation protects from DSS-colitis (Chang, Medzhitov et al., 2015 or Venkataraman, Puimood et al., 2000, 2003). Whether this has to do with facility-specific differences in microbiota or the precise timing of DSS treatment is not clear. Regardless of this, several key controls in regard to this set of findings are lacking. Additionally, while bone marrow chimeric studies support the conclusion that NLRP1 in both hematopoietic and non-hematopoietic compartments contributes to the colonic pathology of DSS treatment, whether in the context of the BM chimera experiments, the same mechanism is at work as in the whole-body knockout mice is not demonstrated, and must be inferred (altered microbiota composition, loss of butyrate). Similarly, whether the CAC phenotypes described in Figure 4 have to do with altered butyrate production and changes in microbial composition is also not demonstrated. Overall, the data provide a potentially interesting series of observations but whether they share the same underlying basis is not clearly developed. An additional concern is that the studies with *Nlrp1*^{-/-} mice are not performed with littermates, whereas the *Nlrp1a*^{Q593P} activating mutation studies are performed with littermates. Thus, disparate sets of data are used to buttress conclusions in related, but different systems. Thus, this manuscript would benefit from a complete set of analyses dissecting the underlying basis for any individual model or system that is described.

Specific points:

1. A general concern with the NLRP1 data is the use of non-littermates. To be fair, the authors do address this issue in the text and reference the recent study of Mamontopoulos, and invoke the possibility that this may be ASC-independent, and subsequently do use littermates for the *Nlrp1a* activating mutant studies. However, the authors should provide evidence/support for the logical leap made by doing this, which is that the underlying mechanism for the NLRP1 phenotype and the *Nlrp1a* phenotype is the same. At least some key phenotyping analyses should be repeated with *Nlrp1*^{+/+}, *Nlrp1*^{+/-} and *Nlrp1*^{-/-} littermate controls (or at least the *Nlrp1*^{+/-} and *Nlrp1*^{-/-} littermates). If the littermate studies do indeed recapitulate the findings of separate genetic breeding (as suggested it could be by the *Nlrp1a*^{Q593P} littermate studies) it would be helpful to have a discussion of this result with respect to the results of the co-housing experiments presented in Figure 2.

2. The conclusion that the Nlrp1a activating mutation phenotype is reverted by IL-18 deficiency seems based on studies in which both IL-1R and IL-18 are ablated. Is it really fair to make this conclusion without comparison of the single Il18^{-/-}Nlrp1a^{Q53P} mice? Could the phenotype not be due to both IL-1 and IL-18? A mutation in NLRC4 in patients has been linked to elevated levels of both IL-1 and IL-18, and both cytokines appear to contribute to the auto-inflammatory pathologies observed in that setting (Canna et al., al 2014).

3. In figure 3, there is a general lack of controls – figure b, c lack vehicle-treated controls subsequently treated with DSS shown side-by-side with the vanco treated cohorts. Similarly, fig 3f, g lack control SCFA-treated animals, such as propionate-treated animals, followed by DSS treatment. These control treated samples are critical for appropriate interpretation of these data. Essentially the confusing thing about these data is that the expectation is that vancomycin treatment and butyrate add-back should have opposite effects – comparison of the vanco and butyrate datasets in Fig 3 indicate similar phenotypes – I realize that the comparison is being made between WT and Nlrp1^{-/-} under the same treatment conditions, but the appropriate vehicle controls really need to be included here. It may be that this is the result of different batches of DSS or timing of treatment, etc, but appropriate experimental cohorts should be generated to perform these studies in order to rigorously interpret the outcome.

4. The rescue of the phenotype by butyrate is a nice finding, but is not clear what the butyrate is doing in this context – is it affecting Tregs (as per Arpaia et al) or is it affecting induction of M2 macrophages in the lamina propria (as per Chang et al?)

5. Regarding the findings of the colitis associated cancer – this is an interesting finding, but whether these findings are directly related to the observations in DSS is not clear – either the CAC study should be worked up as a separate study, or further analysis of the microbial composition and butyrate treatment should be performed in this context to clarify the relationship between hyperproliferative response of the colonic epithelium of these mice and failure to repair in the context of acute DSS.

6. The human NLRP1 expression data are interesting, but I suspect that a large number of inflammatory genes correlate positively with NLRP1 expression in human UC samples. Presumably the increase in IFN γ is linked in some functional way to the IL-18 observation described in Fig 6? Is there a similar correlation with IL-18 levels in these patients? And can the authors stratify UC patient data with regard to levels of NLRP1/IL-18/IFN γ , with some predictive information regarding whether the highest levels of NLRP1/IL-18/gamma might be amenable to anti-IL-18 therapy, which has recently been described in the setting of activating mutations of NLRC4?

Minor:

The title of figure 1 legend does not match the data or the discussion of the data in this results section. Both the hematopoietic and non-hematopoietic compartments are clearly involved, as noted in the text of this results section. The legend should say this as well.

The title of SFig3 legend is confusing: Depletion of Gram-positive bacteria removes IFN γ increase from NLRP1 deficient mice.

Fig 5d – please show representative colon of untreated colons from each of the genotypes

Fig 6d. same as Fig 5d – please show representative untreated colon histology. also why is Fig 6d swiss roll histology but Fig 5d is cross section of colon? Impossible to directly compare data in Figs 5 and 6

In general the resolution of the histology seems a bit sub-optimal – can this be improved, and key features of the histopath highlighted in the figure and figure legend?

There is a large amount of supplemental immunophenotyping. Given that there is not actually a physical space constraint on the figures – I think some of these data would benefit being moved to the main text.

Reviewer #2 (NLRP3, inflammasome, TLR)(Remarks to the Author):

In the manuscript entitled “NLRP1 exacerbates inflammatory bowel disease through IL-18, restricting butyrate producing Clostridiales”, the authors show that the NLRP1 inflammasome promotes DSS-induced colitis via IL-18, independent of IL1b and modulates the abundance of butyrate producing Clostridiales. The manuscript is well written and the finding in human samples is pertinent and adds to the translational value of the report. However the data as presented does not support all of the conclusions made by the authors. Therefore, we recommend that the authors address the following points to improve the manuscript:

Major concerns:

The microbiome analysis of the NLRP1 mutant and WT mice are interesting, but the authors need to provide more detailed information. For example, does the Nlrp1 mutation cause changes in bacterial diversity and microbiome composition?

For the cohousing study, the cohoused WT mice have attenuated disease. The authors use a Venn diagram to illustrate the microbiota distribution between single housed Nlrp1^{-/-}, and cohoused WT and Nlrp1^{-/-} mice. But an interesting observation in this study is the difference between single housed WT and cohoused WT. The author should include the single housed WT in the Venn diagram. Importantly, the authors should show the specific strains that were increased after cohousing with Nlrp1^{-/-} mice compared to the single housed WT in a bar graph. As this is the main data that concludes that Clostridiales are the beneficial strains that were passed from Nlrp1^{-/-} mice to the WT mice, this data is critical.

In addition to Clostridiales, vancomycin treatment ablates other gut gram-positive bacteria. Thus, the authors should change “Clostridiales depletion” in line 188, to “Vancomycin treatment”. Although it is appreciated that this may be difficult, it would also be more convincing if the authors isolated specific Clostridiales strains and transfer them back into the depleted mice to show that Clostridiales are necessary and sufficient to protect of Nlrp1^{-/-} mice against colitis. Furthermore, the authors need to include control mice receiving DSS but not vancomycin to conclude that vancomycin treatment ablates the Nlrp1 phenotype.

In Figure 3, the authors claim that butyrate treatment can protect WT mice from DSS-induced colitis compared to butyrate treated Nlrp1^{-/-} mice. However, the authors need to include control mice receiving DSS but not butyrate to conclude that butyrate protects mice from DSS colitis.

In figure 4, the H&E and PCNA staining of colon sections need to be quantified.

In supplemental figure 5, the authors showed an increase in the total number of CD4⁺ T cells in the colon lamina propria of the Il1r^{-/-}Il18^{-/-} Nlrp1^{Q593P/Q593P} mice. Although the frequency of IFN γ and IL17A producing cells remain unchanged in the CD4⁺ T cell population, there are more IFN γ and

IL17A CD4⁺ producing cells in the Il1r-/-Il18-/- Nlrp1aQ593P/Q593P mice. This seems contradictory to the authors' conclusion and therefore needs clarification.

Reviewer #1 (PRR, inflammasome, bacterial infection)(Remarks to the Author):

1. A general concern with the NLRP1 data is the use of non-littermates. To be fair, the authors do address this issue in the text and reference the recent study of Mamontopoulos, and invoke the possibility that this may be ASC-independent, and subsequently do use littermates for the Nlrp1a activating mutant studies. However, the authors should provide evidence/support for the logical leap made by doing this, which is that the underlying mechanism for the NLRP1 phenotype and the Nlrp1a phenotype is the same. At least some key phenotyping analyses should be repeated with Nlrp1+/+, Nlrp1+/- and Nlrp1-/- littermate controls (or at least the Nlrp1+/- and Nlrp1-/- littermates). If the littermate studies do indeed recapitulate the findings of separate genetic breeding (as suggested it could be by the Nlrp1aQ593P littermate studies) it would be helpful to have a discussion of this result with respect to the results of the co-housing experiments presented in Figure 2.

We thank the referee for emphasising this point, and agree that data from littermate controls is critical to confirm our findings. Therefore, we intercrossed Nlrp1+/- mice, and performed 16S microbiome analysis of stool from WT and NLRP1-/- littermates that were housed individually for 6 weeks after weaning. This confirms our findings with Nlrp1a mutant mice, that the DSS-colitis phenotype is associated with genotype, not the breeding strategy (**New Figure 2a**).

2. The conclusion that the Nlrp1a activating mutation phenotype is reverted by IL-18 deficiency seems based on studies in which both IL-1R and IL-18 are ablated. Is it really fair to make this conclusion without comparison of the single Il18-/-Nlrp1aQ53P mice? Could the phenotype not be due to both IL-1 and IL-18? A mutation in NLRRC4 in patients has been linked to elevated levels of both IL-1 and IL-18, and both cytokines appear to contribute to the auto-inflammatory pathologies observed in that setting (Canna et al., al 2014).

We agree that this is a caveat to our study. Unfortunately, the single Il18-/-Nlrp1aQ593P mice have a spontaneous inflammatory phenotype and die prematurely at approximately 7 weeks of age (Masters et al Immunity 2012). For this reason we are unable to examine their phenotype in the mouse model of DSS induced colitis. Nevertheless, our studies do show that Il1R-/-Nlrp1aQ593P mice display profoundly exaggerated DSS induced colitis, compared to Il1R-/-Il18-/-Nlrp1aQ53P mice, which definitively implicates a role for IL-18. It does not, however, rule out a role for IL-1b, and we have therefore moderated our discussion appropriately (**Line 299**).

3. In figure 3, there is a general lack of controls – figure b, c lack vehicle-treated controls subsequently treated with DSS shown side-by-side with the vanco treated cohorts. Similarly, fig 3f, g lack control SCFA-treated animals, such as proprionate-treated animals, followed by DSS treatment. These control treated samples are critical for appropriate interpretation of these data. Essentially the confusing thing about these data is that the expectation is that vancomycin treatment and butyrate add-back should have opposite effects – comparison of the vanco and butyrate datasets in Fig 3 indicate similar phenotypes – I realize that the comparison is being made between WT and Nlrp1-/- under the same treatment conditions, but the appropriate vehicle controls really need to be included here. It may be that this is the result of different batches of DSS or timing of treatment, etc, but appropriate experimental cohorts should be generated to perform these studies in order to rigorously interpret the outcome.

These important controls are now included. While it is clear that butyrate treatment does protect the mice (**New Figure 3F**), the vancomycin treatment had a less significant effect (**New Figure 3B and 3C**). We suggest that the reason for this is that vancomycin treatment does not specifically target beneficial, butyrate producing Clostridiales, and is instead a rather broad spectrum gram-positive antibiotic with both protective and deleterious effects in the mouse model of DSS-colitis. Nevertheless the vancomycin treatment did resolve the difference between WT and NLRP1-/- mice in this model.

4. The rescue of the phenotype by butyrate is a nice finding, but is not clear what the butyrate is doing in this context – is it affecting Tregs (as per Arpaia et al) or is it affecting induction of M2 macrophages in the lamina propria (as per Chang et al?)

Our FACS analysis did not reveal a difference in the number of Tregs or M2 macrophages in the lamina propria. To examine a specific mechanism of butyrate as a histone deacetylase inhibitor in Tregs or M2 macrophages we performed FACS analysis of acetyl-histone-H3 using intraepithelial and lamina propria cells. Unfortunately, this analysis did not reveal any differences between WT and Nlrp1-/- mice (**Referee Figure 1**). Therefore, at this time, we have been unable to document the exact mechanism for butyrate in the Nlrp1 deficient mice. Butyrate is a broadly active molecule and new roles are being established regularly, so if there is another specific mechanism that the referee would like us to address we would be happy to do so.

Referee Figure 1. FACS analysis of Acetyl-Histone H3. Intraepithelial or lamina propria cells were collected from the colons of mice, then stained for M1 and M2 macrophages, CD4 T cells and Tregs. No significant differences were observed between WT (Blue) and Nlrp1^{-/-} (Red) mice.

5. Regarding the findings of the colitis associated cancer – this is an interesting finding, but whether these findings are directly related to the observations in DSS is not clear – either the CAC study should be worked up as a separate study, or further analysis of the microbial composition and butyrate treatment should be performed in this context to clarify the relationship between hyperproliferative response of the colonic epithelium of these mice and failure to repair in the context of acute DSS.

To address this issue, we performed the CAC model for WT and Nlrp1^{-/-} mice with and without butyrate treatment. Unfortunately, in this particular experiment the control mice did not develop significant dysplasia, perhaps indicating some technical issues. Although there was a trend towards protection for the Nlrp1^{-/-} mice, and less difference when treated with butyrate, this experiment would need to be repeated for certainty. Therefore, given the significant time required to perform this analysis (3 months experiment + 3-6 months breeding time), we have taken the referee's suggestion to work up this data as a separate study, and have removed it from this submission.

6. The human NLRP1 expression data are interesting, but I suspect that a large number of inflammatory genes correlate positively with NLRP1 expression in human UC samples. Presumably the increase in IFN γ is linked in some functional way to the IL-18 observation described in Fig 6? Is there a similar correlation with IL-18 levels in these patients? And can the authors stratify UC patient data with regard to levels of NLRP1/IL-18/IFN γ , with some predictive information regarding whether the highest levels of NLRP1/IL-18/gamma might be amenable to anti-IL-18 therapy, which has recently been described in the setting of activating mutations of NLRC4?

Initially we were concerned that, as the referee suggests, many inflammatory cytokines would correlate to the expression of NLRP1 in human UC samples. However this is not the case, as IL-18 mRNA expression for example, does not correlate to NLRP1 expression. This is in agreement with the mechanism by which NLRP1 activates IL-18, which is entirely post-transcriptional, with cleavage of pro-IL-18 protein via the inflammasome. In order to determine if cleaved, active IL-18 protein expression correlates with NLRP1 expression in patients with UC we would need to obtain paired samples for western blot, and unfortunately we do not currently have access to this resource. The predictive power of NLRP1 and IFN γ expression in patients with UC could be established in the future as clinical trials to neutralize IFN γ and IL-18 are planned or have commenced.

Minor:

The title of figure 1 legend does not match the data or the discussion of the data in this results section. Both the hematopoietic and non-hematopoietic compartments are clearly involved, as noted in the text of this results section. The legend should say this as well.

The title of this figure has now been changed (**Line 669**).

The title of SFig3 legend is confusing: Depletion of Gram-positive bacteria removes IFN γ increase from NLRP1 deficient mice.

The title of this figure has now been changed.

Fig 5d – please show representative colon of untreated colons from each of the genotypes

These control images are now included (**New Figure 4d**).

Fig 6d. same as Fig 5d – please show representative untreated colon histology. also why is Fig 6d swiss roll histology but Fig 5d is cross section of colon? Impossible to directly compare data in Figs 5 and 6

These control images are now included, all swiss rolls (**New Figure 5d**).

In general the resolution of the histology seems a bit sub-optimal – can this be improved, and key features of the histopath highlighted in the figure and figure legend?

Full resolution images have now been supplied. If key features of the histopathology are still not clear, we will be happy to highlight them in the figure and figure legend.

There is a large amount of supplemental immunophenotyping. Given that there is not actually a physical space constraint on the figures – I think some of these data would benefit being moved to the main text.

We will discuss with the editors about how best to place some of the phenotyping data in with the main text.

Reviewer #2 (NLRP3, inflammasome, TLR)(Remarks to the Author):

The microbiome analysis of the NLRP1 mutant and WT mice are interesting, but the authors need to provide more detailed information. For example, does the Nlrp1 mutation cause changes in bacterial diversity and microbiome composition?

This is an important point. There was no difference in microbial community composition between WT and Nlrp1^{-/-} mice, as determined by RDA, CCA and ADONIS. Diversity and richness were not associated with genotype (multiple regression analysis, corrected for gender P > 0.1). This is now mentioned in the manuscript (**Lines 167-170**).

For the cohousing study, the cohoused WT mice have attenuated disease. The authors use a Venn diagram to illustrate the microbiota distribution between single housed Nlrp1^{-/-}, and cohoused WT and Nlrp1^{-/-} mice. But an interesting observation in this study is the difference between single housed WT and cohoused WT. The author should include the single housed WT in the Venn diagram. Importantly, the authors should show the specific strains that were increased after cohousing with Nlrp1^{-/-} mice compared to the single housed WT in a bar graph. As this is the main data that concludes that Clostridiales are the beneficial strains that were passed from Nlrp1^{-/-} mice to the WT mice, this data is critical.

We agree that the single and co-housed WT data is important to present as a bar graph. Therefore, this data is now included instead of the Venn diagram, where each OTU can be observed individually (**New Figure 2e**).

In addition to Clostridiales, vancomycin treatment ablates other gut gram-positive bacteria. Thus, the authors should change "Clostridiales depletion" in line 188, to "Vancomycin treatment". Although it is appreciated that this may be difficult, it would also be more convincing if the authors isolated specific Clostridiales strains and transfer them back into the depleted mice to show that Clostridiales are necessary and sufficient to protect of Nlrp1^{-/-} mice against colitis. Furthermore, the authors need to include control mice receiving DSS but not vancomycin to conclude that vancomycin treatment ablates the Nlrp1 phenotype.

We have changed the text describing vancomycin treatment, as suggested (**Line 186 and 694**). Unfortunately, the culture and transfer of specific Clostridiales strains identified here is extremely challenging, and currently beyond our ability to complete for the current manuscript. Appropriate non-vancomycin treated controls are now included (**New Figure 3b**).

In Figure 3, the authors claim that butyrate treatment can protect WT mice from DSS-induced colitis compared to butyrate treated Nlrp1^{-/-} mice. However, the authors need to include control mice receiving DSS but not butyrate to conclude that butyrate protects mice from DSS colitis.

We have now included this important control (**New Figure 3F**).

In figure 4, the H&E and PCNA staining of colon sections need to be quantified.

We quantified this data, however based on the comments of referee 1, have removed this data from the manuscript at this time.

In supplemental figure 5, the authors showed an increase in the total number of CD4⁺ T cells in the colon lamina propria of the Il1r^{-/-}/Il18^{-/-} Nlrp1aQ593P/Q593P mice. Although the frequency of IFN γ and IL17A producing cells remain unchanged in the CD4⁺ T cell population, there are more IFN γ and IL17A CD4⁺ producing cells in the Il1r^{-/-}/Il18^{-/-} Nlrp1aQ593P/Q593P mice. This seems contradictory to the authors' conclusion and therefore needs clarification.

We have amended our discussion of this result (**Line 257**). Although there was a trend towards an increased absolute number of CD4⁺ T cells in the colon of Il1r^{-/-}/Il18^{-/-} Nlrp1aQ593P/Q593P mice, this was not statistically significant, and as there is no difference in the percentage of IFN γ and IL17A CD4⁺ producing cells, this is unlikely to have an effect on phenotype of these mice. In contrast, Il1r^{-/-}Nlrp1aQ593P/Q593P mice have a statistically significant increase in the number of CD4⁺ T cells in the colon, and the percentage of IFN γ CD4⁺ producing cells is also significantly increased, so there is a marked cumulative effect.

Reviewer #3

It is possible that the abundance of Clostridiales is actually altered in response to the changes of other bacterial species that could be a primary target of NLRP1. Use of gnotobiotic mice specifically inhabited by Clostridiales should be useful in the future study.

We agree that this approach would be useful in future studies. As we replied to referee #2, the culture and transfer of specific Clostridiales strains identified here is extremely challenging, and currently beyond our ability to complete for the current manuscript.

REVIEWERS' COMMENTS:

Reviewer #1 (Remarks to the Author):

The authors have largely addressed the concerns raised in the previous submission with new experiments where possible or text changes where needed in the case where new experiments were not possible or the data were not definitive. I appreciate the effort the authors have made in improving the manuscript. This study will make an important contribution to our understanding of NLRP1-microbiota interactions.

Minor point:

The authors indicated that they moderated the discussion of IL-1/IL-18 to indicate that this study has not definitively ruled out a role for IL-1. However, in reading the part of the discussion that is referenced (line 299) I am not sure that the discussion explicitly talks about this issue. The text in the response to reviewer comments is indeed quite thorough and sufficiently explanatory. The text in the actual manuscript in line 299 is very cursory, and just states "We have shown that IL-18 is involved in the exacerbation of DSS-colitis after activation of NLRP1." The authors need to provide the same discussion in the manuscript that they do in this very nice and thorough response to the reviewer comment, noting the issues with the single Il18^{-/-}Nlrp1Q593P mutation.

Reviewer #3 (Remarks to the Author):

None

REVIEWERS' COMMENTS:

Reviewer #1 (Remarks to the Author):

The authors have largely addressed the concerns raised in the previous submission with new experiments where possible or text changes where needed in the case where new experiments were not possible or the data were not definitive. I appreciate the effort the authors have made in improving the manuscript. This study will make an important contribution to our understanding of NLRP1-microbiota interactions.

Minor point:

The authors indicated that they moderated the discussion of IL-1/IL-18 to indicate that this study has not definitively ruled out a role for IL-1. However, in reading the part of the discussion that is referenced (line 299) I am not sure that the discussion explicitly talks about this issue. The text in the response to reviewer comments is indeed quite thorough and sufficiently explanatory. The text in the actual manuscript in line 299 is very cursory, and just states "We have shown that IL-18 is involved in the exacerbation of DSS-colitis after activation of NLRP1." The authors need to provide the same discussion in the manuscript that they do in this very nice and thorough response to the reviewer comment, noting the issues with the single Il18^{-/-}Nlrp1^{Q593P} mutation.

Response: We have now added this additional detail into the main text lines 564-568: Although genetic deletion of IL-1R did not rule out a contribution from IL-1b downstream of Nlrp1 in DSS-colitis, it did implicate IL-18 in this process. Unfortunately, *Il-18^{-/-}Nlrp1a^{Q593P/Q593P}* mice have a spontaneous inflammatory phenotype and die prematurely at approximately 7 weeks of age. For this reason we are unable to examine their phenotype in the mouse model of DSS induced colitis, and instead we genetically deleted the gene encoding IL-18 on the *Il-1r^{-/-}Nlrp1a^{Q593P/Q593P}* background.

Reviewer #3 (Remarks to the Author):

None

Response: None required